# Application of a Drone Magnetometer System to Military Mine Detection in the Demilitarized Zone

**DOI:** 10.3390/s21093175

**Published:** 2021-05-03

**Authors:** Lee-Sun Yoo, Jung-Han Lee, Yong-Kuk Lee, Seom-Kyu Jung, Yosoon Choi

**Affiliations:** 1Korea Institute Ocean Science & Technology, Busan 49111, Korea; yools@kiost.ac.kr (L.-S.Y.); leejunghan@kiost.ac.kr (J.-H.L.); yklee@kiost.ac.kr (Y.-K.L.); skjung@kiost.ac.kr (S.-K.J.); 2Department of Energy Resources Engineering, Pukyong National University, Busan 48513, Korea; 3Department of Electronic Engineering, Sogang University, Seoul 04107, Korea

**Keywords:** drone, landmine, magnetometer, demilitarized zone

## Abstract

We propose a magnetometer system fitted on an unmanned aerial vehicle (UAV, or drone) and a data-processing method for detecting metal antipersonnel landmines (M16) in the demilitarized zone (DMZ) in Korea, which is an undeveloped natural environment. The performance of the laser altimeter was improved so that the drone could fly at a low and stable altitude, even in a natural environment with dust and bushes, and a magnetometer was installed on a pendulum to minimize the effects of magnetic noise and vibration from the drone. At a flight altitude of 1 m, the criterion for M16 is 5 nT. Simple low-pass filtering eliminates magnetic swing noise due to pendulum motion, and the moving average method eliminates changes related to the heading of the magnetometer. Magnetic exploration was conducted in an actual mine-removal area near the DMZ in Korea, with nine magnetic anomalies of more than 5 nT detected and a variety of metallic substances found within a 1-m radius of each detection site. The proposed UAV-based landmine detection system is expected to reduce risk to detection personnel and shorten the landmine-detection period by providing accurate scientific information about the detection area prior to military landmine-detection efforts.

## 1. Introduction

The demilitarized zone (DMZ) was established in 1953 to prevent armed conflict between the two Koreas, and covers an area 4 km in width, with 2 km on each side of the military demarcation line (MDL). The civilian control line (CCL), located 5–20 km south of the southern limit line, was also established for the protection and security of military facilities (Figure 1). Due to restrictions on development for military purposes, the natural environments of the DMZ and civilian control zone of Korea have been preserved in a natural state. According to an ecological survey around the DMZ, preservation is essential due to the exceptional biodiversity and high ecological value of this area [1,2,3]. In the future, this area is to be transformed into a global ecological and peace zone.

The DMZ and neighboring areas have the highest concentrations of landmines per unit area in the world. Unofficial figures state that a million mines have been set in the DMZ, which could require a million years for full removal [4]. The landmines have not yet been removed, and the border zone within the civilian control line has been opened to the private sector. To prevent risk to civilians, a step-by-step method for eliminating landmines should be established. Therefore, landmines around the DMZ must be investigated quickly, while preserving the natural environment.

### 1.1. Unmanned Aerial Vehicle (UAV) System for Unexploded Ordnance (UXO) Detection

Metal detectors and ground-penetrating radar (GPR) are non-destructive mine-detection methods that are commonly used by the military. Currently, the military conducts detection operations without prior information about the detection area, which leads to long working hours. It causes work fatigue and reduces the operator’s ability to detect land mines, raising the high risk. To avoid these issues, research is underway to detect landmines by mounting several sensors on an unmanned aerial vehicle (UAV). The advantage of UAVs is that large areas can be economically investigated in a short time. 

In recent years, many studies have been conducted to use UAVs for various purposes, such as safety inspection and facility management in construction sites [5,6,7,8,9], crop monitoring in agricultural fields [10,11,12,13], environmental monitoring [14,15], wildlife monitoring [16,17,18], mineral exportation and operation management in mining sites [19,20] and wireless communication support [21]. In addition, Kanellakis and Nikolakopoulos [22] reviewed visual sensors and computer vision techniques for UAVs. These studies have shown the advantage that UAVs can economically and quickly perform investigations of large areas that are difficult or dangerous for humans to access. However, most of studies used visual sensors that are not suitable for detecting buried landmines. Therefore, it is essential to select a sensor this is suitable for detecting buried landmines while being able to mount on a UAV.

Experiments have been conducted for detection of plastic mines using ambient temperature differences [23]. However, this system cannot be used in areas with numerous bushes and buried mines. GPR has the advantage of being able to obtain images of multiple substances buried underground by radiating microwaves and receiving reflected microwave signals [23,24,25,26]. Landmine detection using GPR is limited in environments with complex underground media and in non-flat environments, as complex scattering obscures the signals reflected from landmines [27]. In addition, the penetration depth is limited in wetlands with high electrical conductivity [28], and lasers may be prevented from reaching the ground in areas with dense vegetation [29].

Magnetic exploration is a traditional physical exploration technique that has been used in various fields, including mineral exploration, wide-area geological investigation, and unexploded ordnance (UXO) detection. Previously, a UAV was equipped with a flux-gate magnetometer for magnetic exploration [30] and, more recently, the possibility of UAV-based methods replacing traditional aeromagnetic surveys has been explored [31,32].

The greatest weakness of high-resolution UXO magnetic exploration using drones is interference from the magnetic field of the drone itself. Most drone systems are non-magnetic, but batteries, drone motors and wires can create strong magnetic interference. High-resolution magnetic exploration with drones requires minimizing the impact of such interference on the magnetometer. To address this weakness, installation of the magnetometer at a specific distance (3–5 m) from the drone has been suggested [33]. However, increasing this distance leads to less stable flight and a greater burden on the drone payload. The flight control strategies of a quadcopter with a cable-suspended payload such as energy-based control (EBC) strategy [34], backstepping and super-twisting integral sliding mode strategies [35] were proposed to keep the stable flight. However, it has only been studied theoretically by simulations in an ideal environment where the altitude is fixed and the altitude does not the change due to the influence of the dust environment. Recently, use of two magnetometers has been proposed to compensate for the magnetic noise from drones [36]. However, installing multiple magnetometers is difficult and requires a larger drone. Larger drones create more magnetic noise and, therefore, suitable drone size is an important consideration [37].

Previous research has suggested that landmines may be detected by attaching one magnetometer to a small UAV (sUAV) [37,38]. The conditions for detecting landmines involve measuring magnetic anomalies and adjusting for the distance between the landmine and the sUAV [39]. To increase the probability of landmine detection, it is essential to maintain a constant altitude, a short distance (1–2 m) from the surface [39]. Research is underway to achieve this goal, but no reliable method has yet been described [37]. Processing of data obtained using UAVs follows existing procedures used for manned and airborne magnetism observations [40]. Traditional methods require multiple steps and complex data processing. New data-processing methods are needed to detect small magnetic anomalies rapidly. The efficacy and practicality of UXO detection using UAVs have not been tested under real-world conditions, as most methods have been tested in limited and controlled environments.

### 1.2. Aim and Scope

The main contribution of this paper is to propose simple data-processing methods to minimize the noise caused by the swing of the hanging sensor, and installation method of the magnetometer itself to improve the mine identification. To validate the proposed methods, this study assesses the practicality of UAV systems for landmine detection by the military near the DMZ. To minimize magnetic noise from the drone and increase the probability of landmine detection, the magnetometer sensor is installed to operate in a pendulum manner, hanging about 50 cm below the landing pole of drone. To maintain constant altitude and stable flight of drone with hanging the magnetometer, an altimeter suitable for the experimental environment was evaluated and selected. A simple data-processing method was developed for detecting small magnetic anomalies. After magnetic field exploration was completed, the military conducted verification of the detected magnetic anomalies.

The remainder of this paper is structured as follows: Section 2 describes the configuration of the newly proposed UAV system, Section 3 introduces the data-processing method used to detect small magnetic anomalies, such as metal antipersonnel mines (M16), Section 4 describes the results of UAV magnetic surveys conducted in actual landmine-detection environments and, finally, Section 5 provides conclusions and directions for future studies.

## 2. Experimental Materials

### 2.1. Drone System

The drone system proposed by the author in the previous study [39] is 1.0 m (L) × 1.0 m (W) × 0.7 m (H) in size, with a total weight of 5 kg (including batteries), and the magnetometer is fixed and installed at the center of the landing pole beneath the drone (Figure 2). The distance of the magnetometer from the center of the drone is about 0.7 m (Figure 2a). The system has a high risk of fall due to collisions between landing poles and vegetation during low altitude flight in lush vegetation areas such as the DMZ. Therefore, the system could not be operated below 1 m altitude for stable flight. 

To solve this problem, this study operated the magnetometer in a pendulum manner, hanging about 50 cm below the landing pole, which allows a reduced distance between the land surface and the magnetometer (Figure 2b). This reduces the risk of collision between landing poles and vegetation and increases the signal-to-noise ratio, thereby increasing the probability of detection and reducing self-interference from the drone. 

The size and specifications of the drones used in this study are as described previously, and the optimal pendulum distance is determined by the fourth difference calculation method [41] when the allowable noise level of the magnetometer is lower than 0.1 nT (peak to peak). The distance from the magnetometer to the center of the drone is about 1.2 m.

### 2.2. Real-Time Kinematic (RTK) System

The drone in this study uses the Here+ RTK global positioning system (GPS) device based on U-blox M8P, which is commonly used in drones with the Pixhawk flight controller (FC). This RTK has centimeter-level accuracy. The drone system is equipped with the HERE+rover and HERE+base antenna system on the ground control station (GCS) to improve location accuracy (Figure 3).

### 2.3. Magnetometer

The fluxgate magnetometer is a three-component vector magnetometer with perpendicular components that measure magnetic fields in all directions. The advantages of this magnetometer include low cost, low weight and low power consumption, making it practical for use mounted on a drone [42,43]. To reduce noise in the magnetic field, we replaced the magnetometer, reducing the sampling rate from 50 to 20 Hz [44]. The magnetometer used in this system is the Model 1540 produced by Applied Physics System (Mountain View, CA, USA), which is relatively cheaper than other commercial magnetometers. The magnetometer range is −65 to 65 µT, with a resolution 0.01 nT and a sampling rate of 20 Hz.

The fourth difference calculation was used to measure magnetic noise from the battery and motor of the rotating-wing drone, a method that is used to determine noise levels in aircraft development. The fourth difference value was acquired within a range of 0.1 nT (peak to peak) [41], with the magnetometer located 1.2 m from the drone.

### 2.4. Radio Frequency (RF) Modem and Wireless Network (WiFi) Transmission

A 900-MHz telemetry radio based on the Pixhawk FC was used to control the drone remotely from the GCS. The drone-mounted mission computer processes (saves and merges) flight information from the drone together with magnetic field data. The GCS monitors real-time merged data transmitted from the mission computer using the WiFi router, and controls real-time magnetometer sensors via WiFi (Figure 4).

### 2.5. Altimeter Selection Experiment

The most commonly used altimeters use ultrasonic and laser sensors. An ultrasonic sensor measures distance by releasing ultrasonic waves and calculating the time until reflected waves are received from the target, while a laser altimeter detects the ground reflection of a transmitted laser signal to measure altitude precisely.

Ultrasonic altimeters are not suitable in areas with dense vegetation [45], as they use long wavelengths that are prone to vegetation interference [46]. Laser altimeters, on the other hand, show excellent performance on flat ground or over dense vegetation [47], but dust disturbed during low-altitude (<2 m) drone flight can interfere with signal reception and thus height calculation [48]. Therefore, it is essential to use a laser altimeter that is not affected by dust generated during flight.

To measure altitude changes in a dusty environment, two types of laser altimeters were mounted on a drone, which was set to an altitude of 2 m and flown automatically. The altitude measured by the LW20 laser altimeter ranged from 0.74 to 3.8 m. The average altitude was 2.02 m and the standard deviation was 0.53 m. SF11c had a range of 1.84 to 2.18 m, with an average altitude of 2.03 m and a standard deviation of 0.05 m (Figure 5). Although it is a laser altimeter, the latter model produced stable altitude readings in an environment where beam width was relatively narrow, and the signal strength was seven times stronger than with the former model (Table 1). Therefore, the SF11c altimeter with the most stable altitude value was selected for this study. 

## 3. Data Processing to Distinguish M16 Land Mines

Magnetic anomalies of landmine arise either from induced magnetization in the present-day geomagnetic field related to magnetic susceptibilities or from remanent magnetization [49]. The goal of the data-processing method proposed in this study was to obtain magnetic field exploration results from the UAV as soon as possible and provide them to the military for landmine-detection operations. However, the results of magnetic exploration can be difficult to interpret due to the variety of magnetic substances present underground. To this end, a data-processing method was established to distinguish magnetic anomalies that are similar to those from M16 mines.

First, the magnetic anomaly generated by an M16 mine was measured by an automatic flight at drone altitudes of up to 3 m above the surface near the survey area.

Total intensity was determined using the X-, Y-, and Z-component directions of the vector magnetometer, with a non-linear filter applied to remove spike noise. A Gaussian low-pass filter was used to remove (Figure 6a) the effect of the pendulum motion of the magnetometer below the drone on the magnetic field [50]. For a three-axis magnetometer, each orthogonal axis has its own error. As a result, the total intensity changes with changes in heading. To remove the influence of the lateral direction, spatial filtering was performed. This process applied a high-frequency pass filter based on the moving average, which involves calculating the average values at the front and rear points of the measurement line and removing them from the measurement values using the moving average mean (Figure 6b):(1)$$D^{\prime }\left(i\right)=D\left(i\right)-\frac{1}{2N}\left[\overset{i-1}{\underset{j=i-N}{\sum }}D\left(j\right)+\overset{i+N}{\underset{j=i+1}{\sum }}D\left(j\right)\;\right]$$
where $D^{\prime }\left(i\right)$ is the processed magnetic field at the i-th station, and D(i) is the magnetic field acquired in the field at the *i*-th station.

When the moving average interval (*N*) was set to 10, the absolute value of the M16 mine signal was greater than 5 nT at an altitude of 1 m (Figure 6c). Therefore, 5 nT should be assumed to be the minimum value for landmines, and values below 5 nT should be ignored. For diurnal variations, the international geomagnetic reference field (IGRF) correction was not applied.

## 4. Results and Discussions

### 4.1. Site Introduction

The study area (Figure 7) is located in Yul-ri, Cheorwon-eup, Cheorwon-gun, Gangwon-do, and is 10 km south of the center of the DMZ. Most minefields remaining in the Cheorwon area are located in forest lowlands with lush vegetation [45]. The study site is rented to civilians for farming, but civilians have been restricted from entering because a M16 was discovered by a civilian in Area A during reclamation work. The survey site can be divided into two areas. In Area A, mine detection has been performed by the military, while Area B is an area in which mine detection is planned. Area A has flat topography and almost no bushes, whereas Area B contains a dense distribution of bushes more than 0.5 m in height. We conducted magnetic exploration of the two survey areas using a mine-detection drone.

### 4.2. Drone Flight

Magnetic survey flights were performed for each area. The study areas were 30 m × 35 m and 10 m × 50 m, respectively. The flight interval and altitude were 1 m and the flight speed was set to 1 m/s for automatic flight. Flights were conducted reliably in Area A, despite large amounts of dust being generated by the drone. The flight altitude averaged 1.09 m and the standard deviation was 0.1 m.

In Zone B, the drone aimed to maintain a distance of 0.5 m above the bushes. Depending on the height of the vegetation, the altitude ranged from 1.0 to 1.5 m during the survey. Flights conducted in Zone B were stable, but the irregular vegetation sometimes caught on the magnetometer hanging from the drone during the flight. The areas with vegetation more than 2 m high on the left and right sides of Area B were outside the detectable range of M16 mines, and thus were excluded from the survey.

### 4.3. Magnetic Survey Results

Total magnetic intensity in Area A (Figure 8a) was in the range of 51,400–52,100 nT, and a strong dipole-type magnetic anomaly was observed in the center of the survey area. In the distribution of total magnetic intensity, no specific magnetic anomalies were identified aside from the strong magnetic anomaly in the center. As a result of applying the proposed data-processing method to classify small-scale magnetic anomalies similar to those of M16 mines, two possible signals were identified (Figure 8b).

Total magnetic intensity in Area B (Figure 8c) was in the range of 50,700–52,200 nT. Although a very complex distribution of magnetic anomalies was observed, it was difficult to identify small-scale magnetic anomalies caused by landmines. However, after applying the proposed data-processing method, seven possible mine-related signals were identified in Area B (Figure 8d).

### 4.4. Anomaly Detection and Military Identification Results

The magnetic anomalies identified in Area A were immediately confirmed by the explosive ordnance disposal (EOD) unit waiting at the site (Figure 9). The sites of the magnetic anomalies and the locations of the identified objects were less than 1 m apart, and at depths of about 20 cm. Signal A-1 in Area A was identified as buried rebar, and A-2 was identified as a shell casing.

The military check of Zone B was conducted one week after UAV monitoring, in accordance with the military’s schedule, and the search results were provided to the author. As a result of the search in Area B, cans were found at the locations of signals B-1 and B-2, shells at B-4, waste motors at B-5, iron pipes at B-6, construction waste at B-7 and a piece of rebar at B-8. To verify the accuracy of the magnetic survey results, verification was performed at points B-3, B-9, B-10 and B-11, which had magnetic anomalies of less than 5 nT, but no objects were found (Table 2). During additional mine exploration by the EOD unit, no additional metallic objects were found in areas A or B.

## 5. Conclusions

The sUAV and magnetometer system were operated in an actual mine-detection area. To increase the likelihood of detecting mines, the magnetometer was installed in a pendulum manner under the drone to reduce the distance between the ground surface and the magnetometer. The laser altimeter was improved to ensure stable drone flight at low altitudes by accurately calculating the altitude, even in a dusty environment and an environment with numerous bushes.

The magnetometer was mounted below the drone in the form of a pendulum to detect M16 mines, which showed magnetic anomaly values of 5 nT or more from an altitude of 1 m. From the results of the magnetic survey conducted in the mine-detection area, magnetic small-scale signals caused by abnormal objects such as mines were difficult to confirm based on their total magnetic field. However, after applying the proposed data-processing method, small-scale magnetic anomalies were clearly observed.

During field operations to confirm the magnetic anomalies, objects that were buried or covered by bushes were identified at all except four points that had magnetic anomaly values of less than 5 nT, and no additional metallic objects were found during subsequent mine detection efforts by the military. The proposed UAV system successfully detected UXO with the same magnetic anomaly signal as a M16 in the military mine search area. This indicates that the suggested system can detect M16. When detecting UXO with small magnetic anomalies, results were obtained quickly using a simple data-processing method. In the future, detection of mines by the military prior to removal operations using a UAV may reduce the risk to detection personnel by identifying possible land mines, and may also reduce the mine-detection time.

However, during drone-based magnetic detection from a low altitude in an environment with lush vegetation or nearby obstacles, collisions may occur. Vision-based simultaneous localization and mapping (SLAM) and flight control strategies [34,35] may overcome this problem. This study confirmed the locations of magnetic anomalies, and did not consider a method for analyzing their depths. The relationship of burial depth with magnetic anomalies should be explored through further studies in the future, and the proposed data-processing method should be applied to other mine-detection technologies.

## Figures and Tables

**Figure 1 sensors-21-03175-f001:**
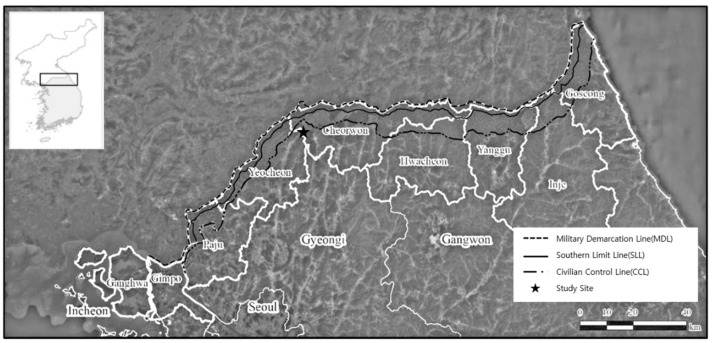
Map of the study site showing the location of the military demarcation line (MDL), southern limit line (SLL), and civilian control line (CCL). The civilian control zone is located between the SLL and CCL. The border line (BL) is the zone within 25 km of the CCL (modified from Lee’s map [2]).

**Figure 2 sensors-21-03175-f002:**
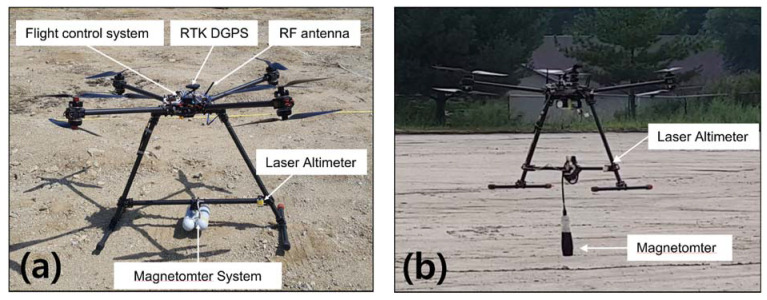
Photograph of the drone-fitted magnetometer system before (**a**) and after (**b**) improvement.

**Figure 3 sensors-21-03175-f003:**
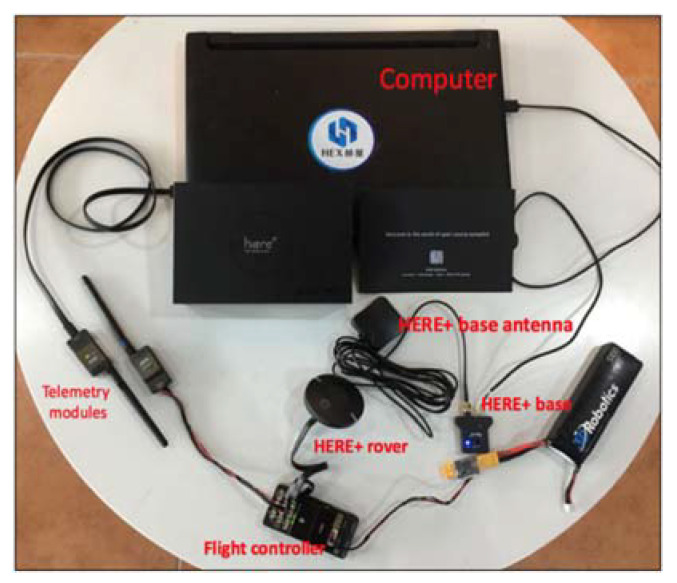
Drone RTK navigation and telemetry systems.

**Figure 4 sensors-21-03175-f004:**
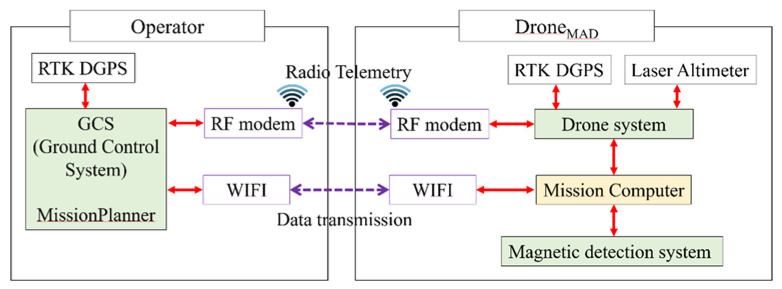
Schematic diagram of the navigation system and magnetic data flow.

**Figure 5 sensors-21-03175-f005:**
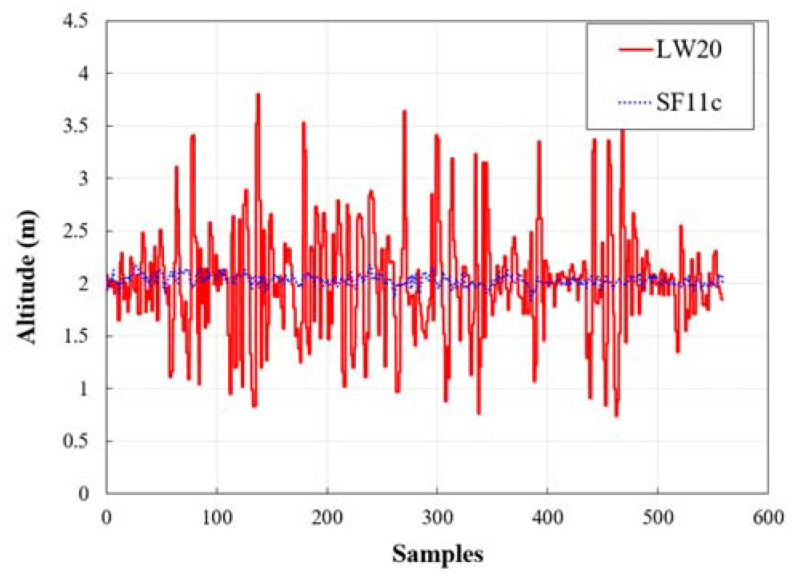
Changes in altitude recorded by altimeters in a dusty environment.

**Figure 6 sensors-21-03175-f006:**
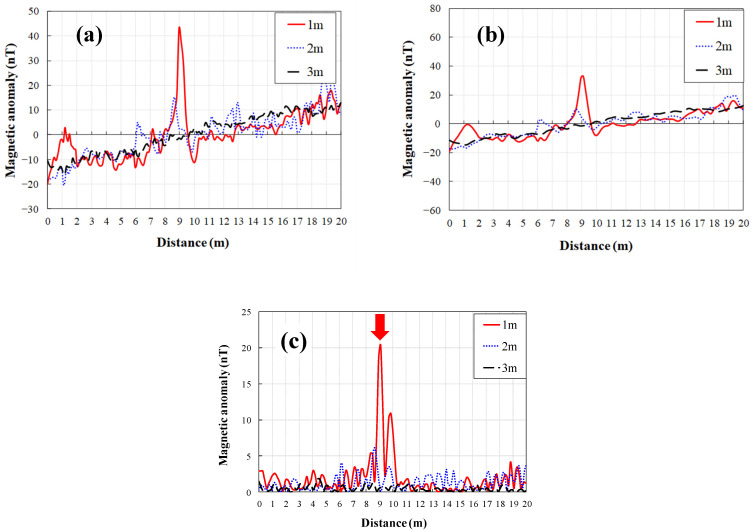
Profile of the proposed data-processing method: (**a**) raw magnetic field along the profile; (**b**) Gaussian-filtered magnetic field using a filter of fc = 10 Hz; (**c**) field detrended using the moving average method. Red Arrow indicate M16 landmine.

**Figure 7 sensors-21-03175-f007:**
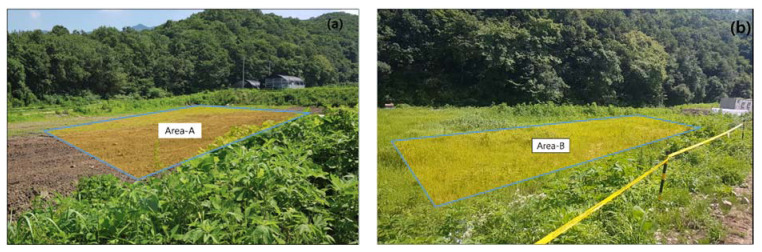
Photograph taken from the southwest of the survey area: (**a**) area where mine removal operations have been completed (30 m × 35 m); (**b**) area scheduled for future mine removal operations (10 m × 50 m).

**Figure 8 sensors-21-03175-f008:**
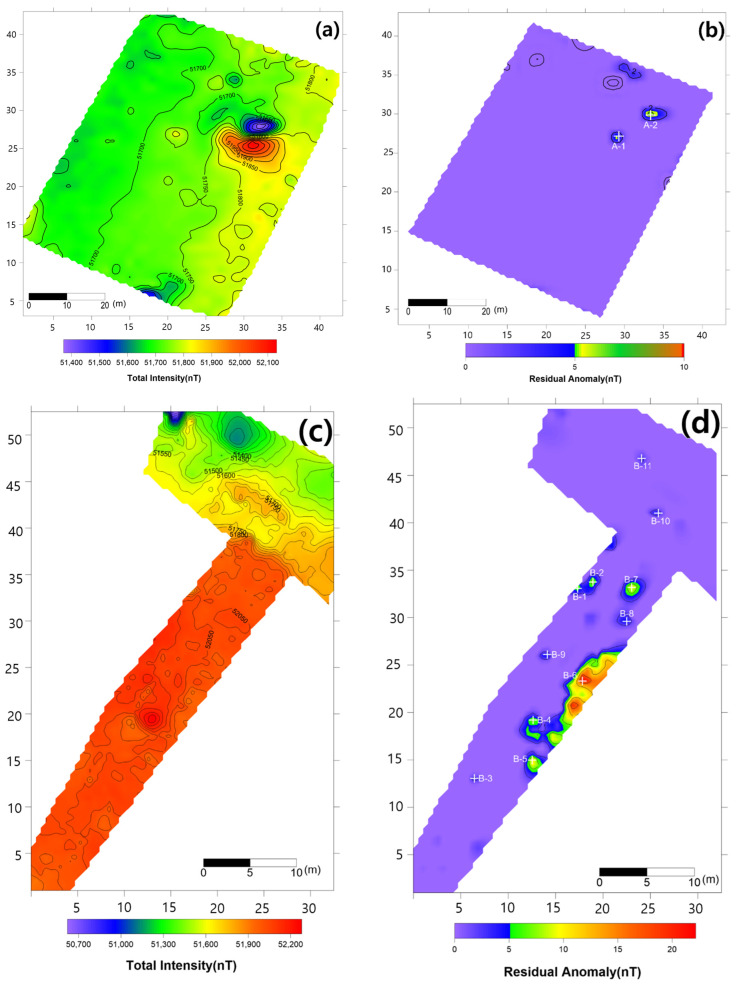
Results of the magnetic survey. (**a**) Total intensity map of Area A; (**b**) map of Area A after application of the proposed data-processing method; (**c**) total intensity map of Area B; (**d**) map of Area B after application of the proposed data-processing method.

**Figure 9 sensors-21-03175-f009:**
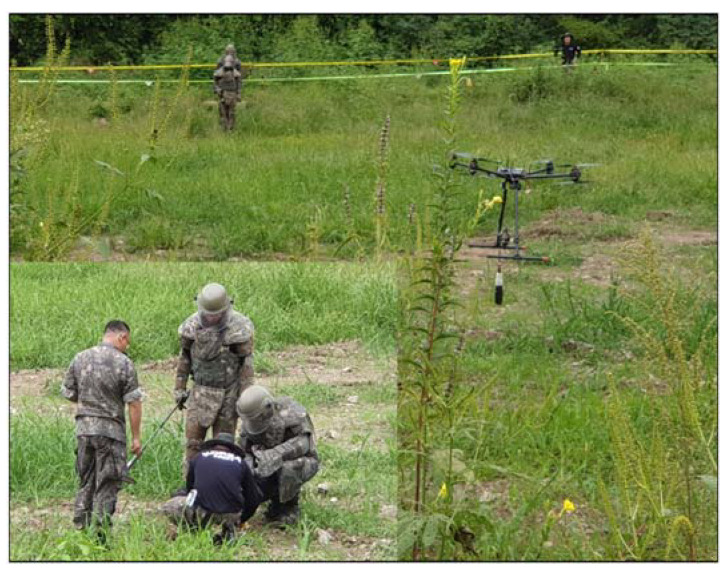
Explosive ordnance disposal (EOD) team operations and abnormal object.

**Table 1 sensors-21-03175-t001:** Specifications of the tested laser altimeters.

Specifications	LW20	SF11c
Type	laser	laser
Resolution	1 cm	1 cm
Accuracy	<10 cm	<10 cm
Power	2 mW	15 mW
Range	<100 m	<120 m
Weight	20 g	35 g
Beam Angle	<0.5°	<0.2°
price	425 USD	390 USD

**Table 2 sensors-21-03175-t002:** Magnetic anomaly-causing objects identified by the EOD.

Areas	Location (m)		Anomaly	Photo	Objects
	x	y	nT		
A-1	28.29	24.15	5.39	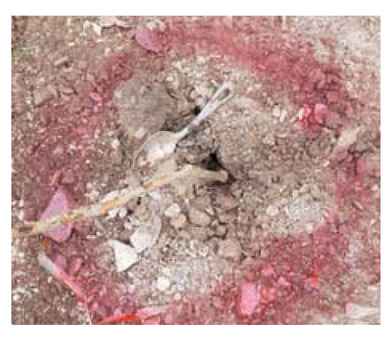	Rebar
A-2	32.37	26.85	6.45	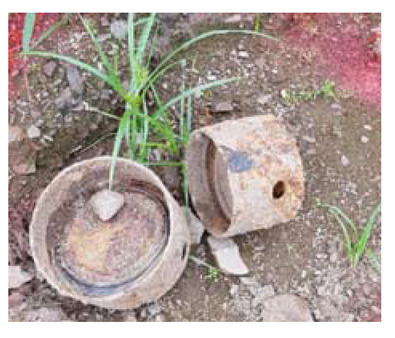	artillery shells
B-1	17.32	32.07	6.21	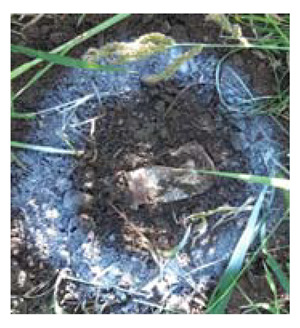	can
B-2	18.94	32.75	8.01	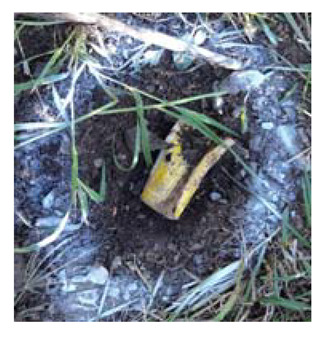	can
B-3	6.43	12.05	2.17	-	none
B-4	12.62	18.2	7.6	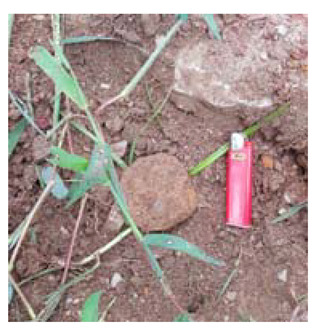	artillery shells
B-5	12.53	13.98	10	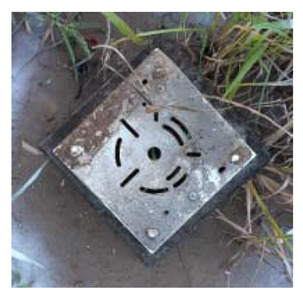	motor
B-6	17.83	22.31	21.2	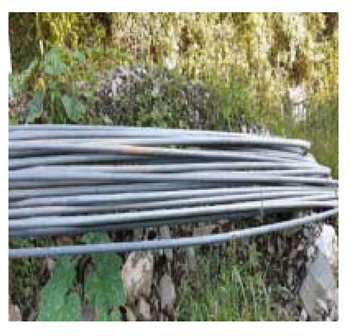	Metal Pipes
B-7	23.02	32.18	9.44	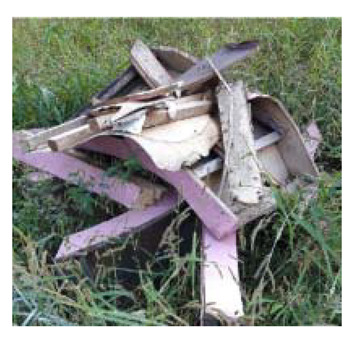	Construction waste
B-8	22.5	28.61	5.13	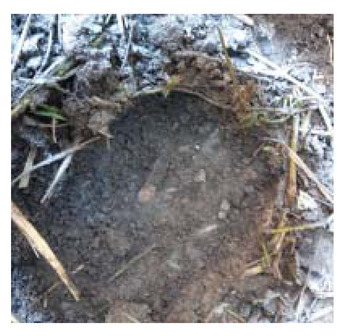	Rebar
B-9	14.12	25.11	3.76	-	none
B-10	25.84	40.01	3.06	-	none
B-11	24.07	45.78	2.36	-	none

## Data Availability

Not applicable.

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
