# Peer review of "Application of a Drone Magnetometer System to Military Mine Detection in the Demilitarized Zone"

_sensors, 2021, doi:10.3390/s21093175_

Round 1
Reviewer 1 Report
Nice publication. Certainly in the scientific and public interest. It reads very clearly and pragmatically, but could be more detailed in some places.
Some comments and suggestions for improvement:
Magnetic Abnomalies:
Please describe better the nature of magnetic Abnomalies. In particular, you should clarify that they are small changes in the existing geomagnetic field caused by the presence of metallic objects. Passages such as "the magnetic field generated by an M16 mine" (line 188) can be misunderstood to suggest that the mine is the original source of the magnetic field itself.
Please clearly highlight the differences in the magnitudes of the regular magnetic field and the observed abnormalities.
UAV Platform:
- Line 123: "However, in the improved system the magnetometer is installed on the landing pole below the drone, which allows a reduced distance between the land surface and the magnetometer."
This is not very clear. What was the problem with the previous version? Collisions of the surface vegetation with the landing frame?
-Did you make any attempts to find the optimal pendulum length? How did you arrive at the length of the pendulum in the improved version (apparently it is only about 20 cm (fig. 2b). How does the flight control system handle it? Did you have to make any changes to it? Would an even longer pendulum length be desirable?
Experimental results:
Please clarify if the results met your expectations. Obviously, no real mines were found in the experiments, but objects causing a similar effect on the magnetic field. Probably that was the expected result. You should make this clearer. In line 283 you write that "and no additional mines were found". Actually I understood that, that no mines were found at all, neither by the UAV nor by human personnel
Further:
-line 44: "Currently, the military conducts detection operations without prior information about the detection area, which carries high risk and leads to long working hours." Although this is of course correct in content, the conflation of lethal threat and sober working hours in the sentence reads somewhat strangely.
Author Response
Dear
Thank you for your valuable comments to improve this manuscript. We revised the original manuscript according to all your comments.

Reviewer 2 Report
This communication manuscript deals with the application of an unmanned aerial vehicle to the detection of landmines using a hanging magnetometer from the vehicle. The UAV travels at low altitude flights and the signals from the magnetometer are treated with a low pass filter for the magnetometer swing and a moving average high-frequency band filter to improve the detection. In my opinion, the paper presents an interesting application of the UAV, that can be useful in minefields.
Some recommendations to improve the paper are:
Despite having an adequate literature review in mine detection, the paper reads as a technology integration only. The scientific contribution must be better emphasized. Please state it explicitly.
According to the description, the sensors used are high-end and possibly the results obtained depend on the quality of the sensor more than on the data filtering/processing algorithms, please comment on this.
Perhaps the most important contribution of this paper is in the data processing to cancel the noise caused by the swing of the hanging sensor and the position of the magnetometer itself to improve the mine identification. In such a way, I suggest including in the literature review control strategies that attenuate the swing of hanging payloads in aerial vehicles, for example:
Energy-Based Control and LMI-Based Control for a Quadrotor Transporting a Payload, Mathematics 7(11), 1090 (2019).
Trajectory tracking for quadrotor UAV transporting cable-suspended payload in wind presence, Trans Inst of Meas and C. 41(5), 1243-1255 (2019).
This in order to complement the literature review.
Figure 5, please include the name and units of the x-axis. Is it time? samples?.
Figure 6, the x-axis is a distance relative to which point? the landing point?
Figure 6. Please include the exact location of the object detected. One assumes that is at the maximum peak, but it might not be the case. The peak spans 1 m or more.
Author Response

(The authors gave the same response as above.)

Round 2
Reviewer 2 Report
The authors have done a great job addressing the reviewers' comments. I am happy with the revised version, I have no further comments.
Author Response
Dear you
Thank you for your valuable comments to improve this manuscript. We revised the manuscript according to all your comments.
We describe UAV applications on 50-60 lines and insert existing literature in references 8, 21, and 22.